# Winter occurrence and spawning characteristics of Pacific herring (*Clupea pallasii*) in Jinhae Bay: An integrated survey using acoustic monitoring, gillnet sampling, and environmental DNA

Euna Yoon[1,2], Yong-Deuk Lee[3], Jeong-Hoon Lee[2], Inyeong Kwon[4], Hyungbeen Lee[2]*

**1** Department of Marine Industry and Maritime Police, Jeju National University, Jeju, Korea, **2** Fisheries Resources Research Center, National Institute of Fisheries Science, Tongyeong, Korea, **3** West Sea Fisheries Research Institute, National Institute of Fisheries Science, Incheon, Korea, **4** Department of Smart Fisheries Resource Management, Chonnam National University, Yeosu, Korea

* hblee00@korea.kr

## Abstract

This study investigated the adult density, timing of migration, and biological characteristics of Pacific herring (*Clupea pallasii*) entering Jinhae Bay, South Korea, in winter, a major spawning ground for the species. A wideband autonomous transceiver (WBAT) was deployed from November 2022 to March 2023, and standardized gillnet surveys and environmental DNA (eDNA) were conducted concurrently. WBAT monitoring showed that *C. pallasii* school signals appeared predominantly from mid-January to mid-February, coinciding with the period during which herring were highly dominant in gillnet catches. Assessment of female reproductive maturity indicated that most individuals were ripe or spent, confirming that this period represented the peak spawning season. eDNA concentrations exhibited a sharp peak in mid-January in both surface and bottom waters and then declined steadily thereafter. Taken together, these results indicate that entry into the spawning ground begins in late-December, peaks in mid-to-late January, and declines toward late-February. By integrating acoustic, catch, and eDNA datasets, this study provides a comprehensive assessment of the timing, density, and spawning ecology of *C. pallasii*, offering evidence-based guidance for future resource management and spawning-ground protection in Jinhae Bay.

## Introduction

Pacific herring (*Clupea pallasii*) are a major commercial fish species in the order Clupeiformes, family Clupeidae. They are broadly distributed across the North Pacific Ocean, including the East Sea and eastern South Sea of South Korea, as well as central and northern Japan, the Bering Sea, and Alaska [1,2]. *C. pallasii* are a major food source for various predators in marine ecosystems and form part of a complex

**Data availability statement:** All relevant data are available from the Figshare repository at https://doi.org/10.6084/m9.figshare.30260569.

**Funding:** This work was supported by the National Institute of Fisheries Science, Korea (grant No. R2025001). The funders provided support for this work and were involved in study design, data collection and analysis, the decision to publish, and preparation of the manuscript.

**Competing interests:** The authors have declared that no competing interests exist.

food web that involves fish, aquatic mammals, and birds [3]. In South Korea, the annual *C. pallasii* catch remained at 10,000 tonnes or higher from the mid-1990s to 2000, decreased to less than 10,000 tonnes by 2005, and then increased again, fluctuating between 20,000 and 30,000 tonnes until 2020 [4].

*C. pallasii* is a cold-water species, most often found at depths of 60–90 m where temperatures of 2–10 °C are maintained. In winter, schooling and coastal movements for overwintering and spawning are observed [2,5]. After movement to the coast, adhesive eggs are laid on reefs and seaweed [5,6]. Given their strong migratory behavior, spawning typically recurs in the same waters each year [6–8]. These ecological characteristics are informative for the evaluation and management of *C. pallasii* resources. In particular, a clearer understanding of the timing of adult coastal entry and the survival of spawned offspring is essential because these processes strongly influence subsequent school size and recruitment [9].

To conserve the commercially and biologically important species, *C. pallasii*, South Korea has implemented a regulation since 2022 that prohibits the landing of individuals smaller than 20 cm [10]. In this context, it is becoming increasingly important to secure scientific data on the time of entry to spawning grounds and the characteristics of spawning ground populations. Recent studies have used multiple approaches, including echosounder, catch surveys, eDNA analysis, ichthyoplankton sampling, and underwater video surveys [11–13]. Among these, wideband autonomous transceivers (WBATs) are advantageous because they enable continuous, fixed-point monitoring of diel and short-term density fluctuations [14]. To ensure accurate species identification, complementary survey methods were employed concurrently, including gillnet catch surveys and eDNA analyses. Catch surveys using gillnets are useful for precisely ascertaining the biological characteristics and distribution of fish through direct sampling. qPCR analysis of eDNA is a technique in which DNA is extracted from organisms in the water to verify the presence or absence of a target species. It has recently been used in various fields, including fish ecology research [15,16]. When different survey methods are combined, they can complement each other to overcome the limitations of each individual method. Studies have been conducted analyzing the correlations between acoustic density, eDNA, and catch volume for Japanese horse mackerel (*Trachurus japonicus*), whiting (*Merlangius merlangus*), and Atlantic cod (*Gadus morhua*), and integrated methods have been demonstrated to be effective [17–19].

The aim of this study is to determine the timing and relative magnitude of *C. pallasii* migration into Jinhae Bay during winter and to characterize adult density and spawning-season patterns. To achieve this, we integrated moored echosounder observations (volume backscattering strength, $S_v$; nautical area scattering coefficient, NASC), standardized gillnet surveys, and species-specific eDNA (qPCR) measurements to track temporal changes in abundance. By evaluating correlations among acoustic indices, catch data, and eDNA concentrations, we provide cross-validation of migration magnitude and spawning activity, supporting evidence-based designation of spawning-ground protection zones and seasonal fishery closures.

## Materials and methods

### Study site and schedule

Temporal variations in *C. pallasii* occurrence and density were monitored for 128 days, from 24 November 2022, to 31 March 2023, using a WBAT (Simrad Kongsberg Maritime AS, Horten, Norway) installed at a depth of 12 m in Jinhae Bay, Gyeongsangnam-do (35° 5.80'N, 128° 41.75'E) (Fig 1A). Jinhae Bay, a semi-enclosed inner bay on Korea's southeastern coast, is characterized by complex topography and serves as a convergence zone for offshore seawater and freshwater inputs. The high terrestrial nutrient input results in elevated primary productivity, making the bay an important spawning and nursery ground for multiple fish and shellfish species. Gillnet catch surveys and eDNA sampling were conducted seven times at two-week intervals between mid-December 2022 and mid-March 2023 at a site located 100 m from the WBAT installation point (35° 5.80'N, 128° 41.80'E; Table 1).

### WBAT installation and analysis of acoustic data

A mooring frame (1 m × 1 m × 0.65 m) was assembled and attached to the main WBAT device for seabed deployment. A 70 kHz (model ES70-7CD; Simrad Kongsberg Maritime AS, Horten, Norway) and a 120 kHz (model ES120-7CD; Simrad Kongsberg Maritime AS, Horten, Norway) transducer were mounted on the upper part of the frame (Fig 1C). During deployment, the transducers were oriented vertically by a diver, and the frame was secured to the seabed using ropes

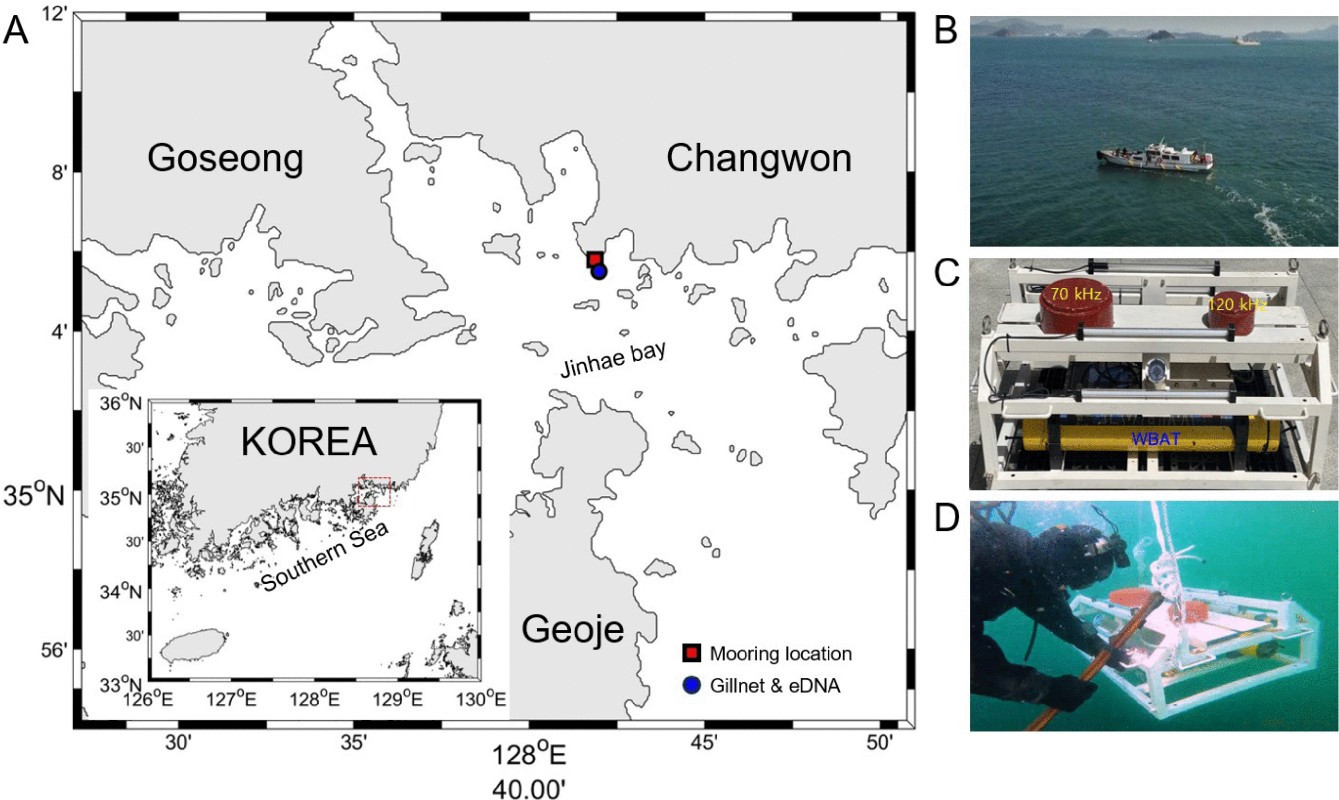

**Fig 1. Study area and survey design in Jinhae Bay, Korea. (A)** The location of the echosounder mooring is shown by the red square, and the station used for eDNA sampling and gillnet catch surveys is indicated by the blue circle. The map was generated using the M_Map toolbox in MATLAB [20]. Panel **(B)** shows a photograph of the vessel used during the survey. Panel **(C)** illustrates the configuration of the moored echosounder equipped with 70 and 120 kHz transducers. Panel **(D)** shows a diver installing the echosounder on the seabed.

**Table 1. Summary of Pacific herring (*Clupea pallasii*) gillnet catch and eDNA surveys conducted in Jinhae Bay, Korea. The table includes survey number, date, sampling location (latitude and longitude), water depth at the survey site, eDNA sampling depths, and filtered seawater volumes for both surface and bottom samples.**

| Station no. | Date | Lat. (°N) | Lon. (°E) | Depth/eDNA depth (m) | Filtered volume (surface/bottom, L) |
|---|---|---|---|---|---|
| 1 | Dec. 15–16, 2022 | 35.097 | 128.697 | 10.2/1, 9 | 0.8/0.8 |
| 2 | Dec. 27–28, 2022 | 35.097 | 128.697 | 12.0/1, 11 | 1.1/1.1 |
| 3 | Jan. 12–13, 2023 | 35.097 | 128.697 | 10.2/1, 9 | 0.8/0.8 |
| 4 | Jan. 30–31, 2023 | 35.097 | 128.697 | 10.0/1, 9 | 1.5/1.9 |
| 5 | Feb. 15–16, 2023 | 35.097 | 128.697 | 12.0/1, 11 | 2.0/2.0 |
| 6 | Feb. 27–28, 2023 | 35.097 | 128.698 | 9.2/1, 8 | 1.4/1.0 |
| 7 | Mar. 15–16, 2023 | 35.097 | 128.698 | 8.8/1, 7 | 1.8/0.5 |

Note: Depth indicates bottom depth at the survey site; eDNA depth indicates sampling depths for water collection; filtered volumes are shown for surface and bottom samples.

tied from the frame rings to anchors to ensure stability against tidal currents. This setup enabled uninterrupted acoustic observations throughout the water column (Fig 1D). Prior to deployment, the WBAT and transducers were calibrated in the seawater acoustic tank at the Fisheries Resources Research Center, National Institute of Fisheries Science (NIFS), using a standard sphere method adapted for the echosounder system [21]. Calibration parameters were obtained using a 38.1 mm diameter tungsten carbide sphere across the frequency bandwidth of each transducer.

The WBAT was configured to transmit an ensemble of 44 pings at 2.3 s every 30 min, with both 70 kHz and 120 kHz frequencies operated in CW mode. Beamwidth was set to 7° for both frequencies. The WBAT battery capacity was 128 Ah, and the maximum feasible transmission rate, considering battery consumption over the survey period, was calculated based on the 44-ping setting. Signal pulse width was set to 0.512 ms for both frequencies, with power levels of 500 W at 70 kHz and 400 W at 120 kHz. Auxiliary data for acoustic signal interpretation were obtained by installing a water temperature sensor (RBR T.D Duet, RBR Ltd., Ottawa, Canada) on the WBAT upper frame, recording measurements at 2-min intervals corresponding to the WBAT operation.

Collected acoustic data were analyzed using the Echoview software platform (Ver. 13.1; Echoview Software Pty. Ltd., Tasmania, Australia) with the virtual echogram method applied to the 70 kHz signal. After removing seabed reflections and near-field interference, data were resampled at 11 ping intervals, and $S_V$ (dB re 1 m$^{-1}$) and NASC (m$^2$ nmi$^{-2}$) were calculated. Detailed analysis of the final extracted signals was conducted in MATLAB (MathWorks, Natick, MA, USA).

## Gillnet catch surveys

From December 2022 to March 2023, gillnet surveys for Pacific herring (*C. pallasii*) were conducted seven times at approximately two-week intervals in Jinhae Bay (Table 1). The gillnets used in the surveys were monofilament nets, with each panel measuring 75 m in floatline length, 3.8 m in height, and a mesh size of 5.15 cm. Two panels were deployed for fish capture, and the nets were arranged in the order of buoy-anchor-net-anchor-buoy. At the start of deployment, the GPS position and water depth were recorded using an echosounder (Plovis 13F, Samyung ENS, Busan, Korea) installed on the survey vessel. The gear was deployed at 18:00 and hauled at 06:00 the following day, after 12 h of submersion (Table 1). The same gear configuration and identical soak duration were applied consistently across all surveys.

All samples obtained from the gillnet surveys were transported in iceboxes to the Fisheries Resources Research Center, National Institute of Fisheries Science. Species identification was performed according to reference [22]. For *C. pallasii*, fork length (FL, to the nearest 0.1 cm) and body weight (BW, to the nearest 0.1 g) were measured for up to 100

individuals, while for other species, total length and body weight were recorded. Gonadal development of female *C. pallasii* was assessed following Murua et al. [23] and classified into five stages: immature, maturing, mature, ripe, and spent. All fish captured using gillnets were found to be dead at the time of collection; therefore, no euthanasia procedures were required. All procedures complied with the guidelines of the National Institute of Fisheries Science, and the protocol was approved by the Institutional Animal Care and Use Committee of the National Institute of Fisheries Science (Protocol No. 2022-NIFS-IACUC-24).

## eDNA survey for species identification

Seawater (2 L each from the surface and bottom layers) was collected in Jinhae Bay using a Niskin bottle (4 L Open-Close bottle, General Oceanics, Miami, Florida, USA) prior to the gillnet deployment. The collected seawater was kept refrigerated at 5 °C and transported to the Aquatic Resources Research Center in Tongyeong-si, Gyeongsangnam-do. In the laboratory, the water samples were filtered through Sterivex filter units (0.45 µm pore size, Millipore, Burlington, MA, USA), and RNAlater (Thermo Fisher Scientific, Waltham, MA, USA) was subsequently injected into the filters as a preservation solution. Variable clogging of filters was observed among samples, and the effective filtered volumes were thereby reduced to 0.5–2.0 L (Table 1). All eDNA concentrations were normalized to copies per milliliter using the sample-specific filtered volume. The preserved filters were stored at –20 °C until DNA extraction. Genomic DNA was extracted from Sterivex filters using the DNeasy Blood and Tissue Kit (Qiagen, Hilden, Germany) following the protocol described by Uchii et al. [24] for environmental DNA samples collected on filter units. The concentration of extracted DNA was measured with a Synergy H1 microplate reader (BioTek, Winooski, Vermont, USA) using the QuantiFluor dsDNA System (Promega, Madison, WI, USA).

For qPCR analysis, a primer set and TaqMan MGB probe targeting the *C. pallasii* cytochrome b region were employed. The sequences of the primers and probe were as developed by Gwak and Nakayama [25]: forward primer 5′-AAA CAA CGG GGC CTA ACA TTC-3′; reverse primer 5′-TAC ACG ACT GAT GCA ACT TGC C-3′; and TaqMan MGB probe 5′-AGC CCT GGC CGC AGA-3′. qPCR reactions were performed with Environmental Master Mix 2.0 (Thermo Fisher Scientific, Waltham, MA, USA) on a QuantStudio 3 Real-Time PCR System (Thermo Fisher Scientific, Waltham, MA, USA), with UNG (Uracil-N-glycosylase) included to prevent carry-over contamination. The Niskin bottle was sterilized prior to sampling, and laboratory negative controls and qPCR no-template controls were included throughout extraction and amplification steps. All reactions were conducted in triplicate. For quantitative calibration, DNA extracted from *C. pallasii* muscle was used to prepare standard solutions at four concentrations to generate qPCR standard curves. These curves enabled absolute quantification and comparison of eDNA copy numbers among samples.

## Statistical analysis

Time-series changes and between-survey differences in water temperature, acoustics, catch, and eDNA data were examined, and correlations and multivariate patterns among variables were analyzed.

To compare water temperature between November 2022 and February 2023, identified as key periods of change, normality was assessed using the Shapiro–Wilk test. Because normality was not satisfied, a Welch's t-test, which does not assume equal variances, was conducted ($\alpha = 0.05$).

Differences in $S_V$ across survey periods were evaluated after testing normality using the Shapiro–Wilk test and homogeneity using Levene's test. A one-way analysis of variance (ANOVA) was then performed, followed by Tukey's HSD post hoc comparisons ($\alpha = 0.05$). For time-series analyses of $S_V$ and NASC, missing WBAT values were linearly interpolated, and short-term outliers were mitigated with a 5-point moving median filter. Local peaks were identified using a prominence-based peak-detection method, and changes in average levels were detected with the PELT algorithm using an RBF kernel cost function.

*C. pallasii* dominance in the gillnet survey was computed as the ratio of the *C. pallasii* catch to the total catch per survey day. Differences among survey time points were evaluated using the Kruskal–Wallis H test ($\alpha = 0.05$). The correlation between dominance and water temperature was assessed using Pearson's correlation.

To comprehensively assess relationships among acoustic, catch, and eDNA data, data preprocessing was performed, followed by correlation analyses and principal component analysis (PCA). To reduce skewness, eDNA and catch values were transformed using $\log_{10}(x + 1)$, and $S_V$ from WBAT was averaged to obtain mean $S_V$ at each sampling point. Missing $S_V$ values were supplemented by linear interpolation, and all variables were standardized to Z-scores. Spearman rank correlations ($\alpha = 0.05$) were used to evaluate associations among variables, and PCA was used to summarize multivariate structure and visualize temporal trajectories of the samples.

## Results

### Monthly acoustic density of *C. pallasii*

The bottom layer water temperature at the WBAT installation site in Jinhae Bay ranged from 6.26 to 17.56°C between mid-November and late-February (Fig 2A). In November, the bottom temperature remained above 15°C, after which it declined steadily, reaching below 7°C in February, before showing a slight increase toward the end of the month (Welch's t-test: $t = 683.38$, $P < 0.05$).

Backscatter signals from fish schools were detected irregularly throughout the monitoring period, with intermittent periods of strong acoustic activity (Fig 2B). From the initial mooring in late-November through December 2022, almost no schooling signals were observed. In contrast, a sharp onset of strong backscatter signals appeared in mid-January 2023, during which the majority of detections were concentrated (ANOVA, $P < 0.05$). After this peak period, signals became intermittent again from February onward, resembling the pattern observed early in the deployment.

The mean volume backscattering strength ($S_V$) ranged from −85.0 to −45.6 dB (Fig 2C). A distinct, high-intensity scattering event persisted for approximately three days (January 14–17, 2023). The nautical area scattering coefficient (NASC), representing the relative density of fish schools, remained ~5 $m^2$ $nmi^{-2}$ during periods without strong signals but increased sharply, reaching a maximum of 34,781 $m^2$ $nmi^{-2}$ in mid-January (Fig 2D). Afterward, NASC values showed a moderate increase through mid-February, culminating in a second peak on February 17, followed by a gradual decline. Time-series volatility analysis revealed that short-duration, spike-like peaks were predominantly clustered in mid-December, mid-January, and mid-February. In addition, change-point detection identified significant shifts in the mean backscatter level in late-December and mid-January (Fig 3), indicating abrupt transitions in school density and aggregation behavior during these periods.

### Monthly catch volume and biological characteristics of *C. pallasii* using gillnets

Throughout the survey period, only two *C. pallasii* individuals were captured in mid-December. However, as water temperature decreased, catch volume increased sharply, reaching a peak of 545 individuals in late-January 2023. From February onward, the catch numbers declined again, and catches were very low in February and mid-March, with only five and two individuals captured, respectively (Fig 4A).

In mid-December 2022, when *C. pallasii* was largely absent, the dominant species was dotted gizzard shad (*Konosirus punctatus*), along with Pacific sardine (*Sardinops sagax*), and yellowfin goby (*Acanthogobius flavimanus*). In contrast, during the four surveys with high *C. pallasii* catch, herring accounted for more than 98% of the total catch, indicating overwhelming dominance (Fig 4B; $H = 6.000$; $P > 0.05$). A significant negative correlation was detected between herring dominance and water temperature (Fig 5; $P < 0.05$), consistent with increased herring presence during colder periods.

The fork length (FL, cm) of *C. pallasii* caught throughout the survey period showed a similar size distribution of 23.3–27.5 cm for females and 23.4–25.4 cm for males (Fig 4C). Most sampled females were in ripe or spent stages, and no

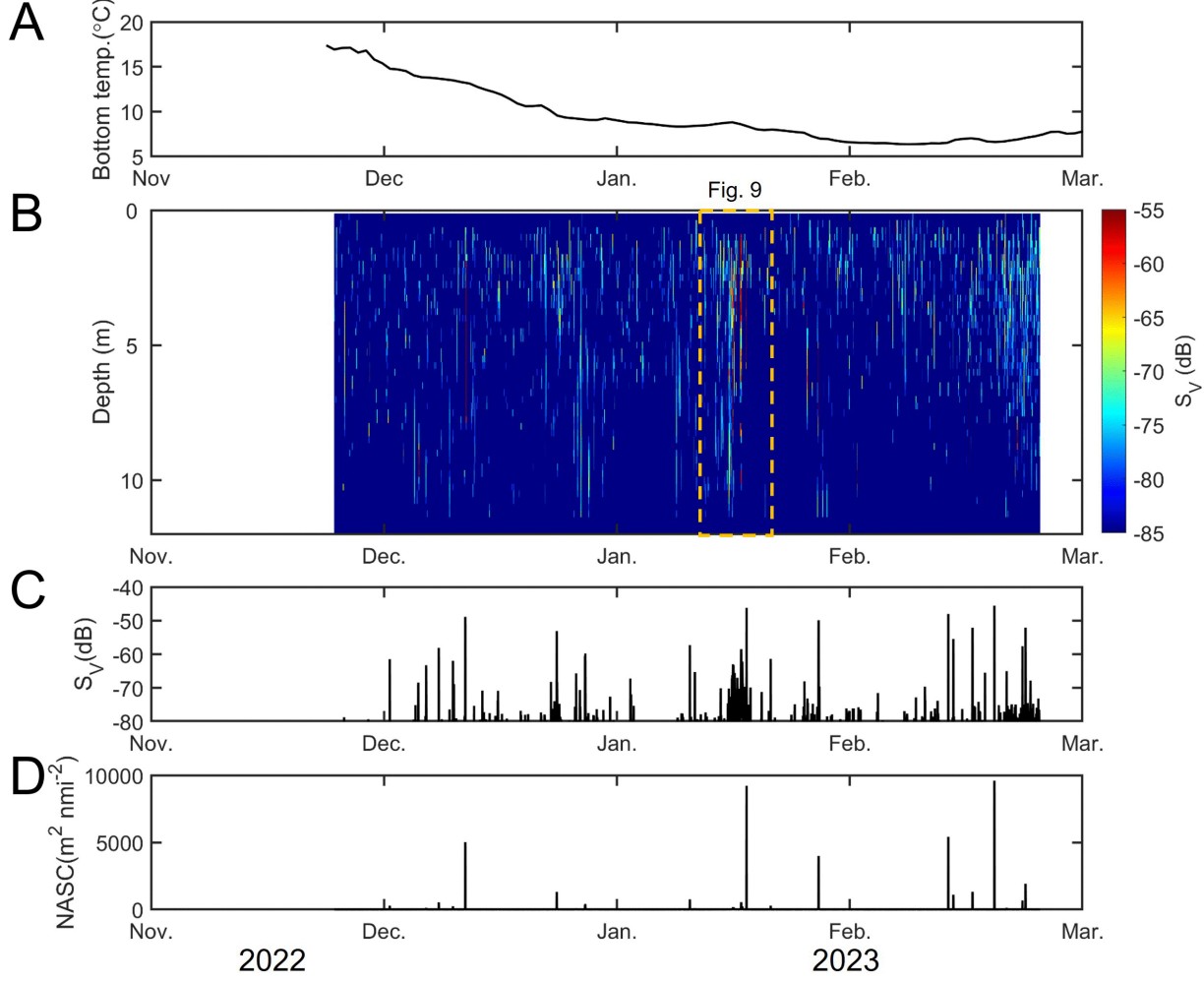

**Fig 2. Temporal variability in bottom water temperature and acoustic backscatter at a mooring site in Jinhae Bay, Korea. (A)** bottom water temperature recorded from November 2022 to March 2023. **(B)** echogram of 70 kHz backscatter. Each data point represents the mean volume back-scattering coefficient for 30-min intervals in a 0.25-m bin. **(C)** mean volume backscattering strength (average $S_v$ in dB); and **(D)** Nautical Area Scattering Coefficient (NASC), representing the integrated backscattering normalized over a horizontal area expressed in $m^2$ $nmi^{-2}$. Strong acoustic peaks were observed in mid-January, corresponding to the main spawning influx of Pacific herring.

immature individuals were observed (Fig 4D). In surveys from late-December, when the *C. pallasii* catch volume began to increase, to mid-February, the proportion of ripe females gradually decreased (84%→56%→28%→13%), whereas the proportion of spent females increased correspondingly (12%→44%→66%→87%). These maturity patterns confirm that adult herring entered Jinhae Bay beginning in late-December and actively participated in spawning.

### *C. pallasii* eDNA concentration using qPCR analysis

Contamination was observed in one negative-control (NC) sample in mid-January, but no contamination occurred in any other field or laboratory NC samples. The quantitative reliability of the qPCR assays was supported by standard curves with a high coefficient of determination ($R^2 = 0.99$), and thus the reported copy numbers are considered robust. *C. pallasii* eDNA was not detected at all in the survey waters in either the surface layer or the bottom layer in December 2022. However, in mid-January 2023, high concentrations of 348,333 copies $mL^{-1}$ in the surface layer

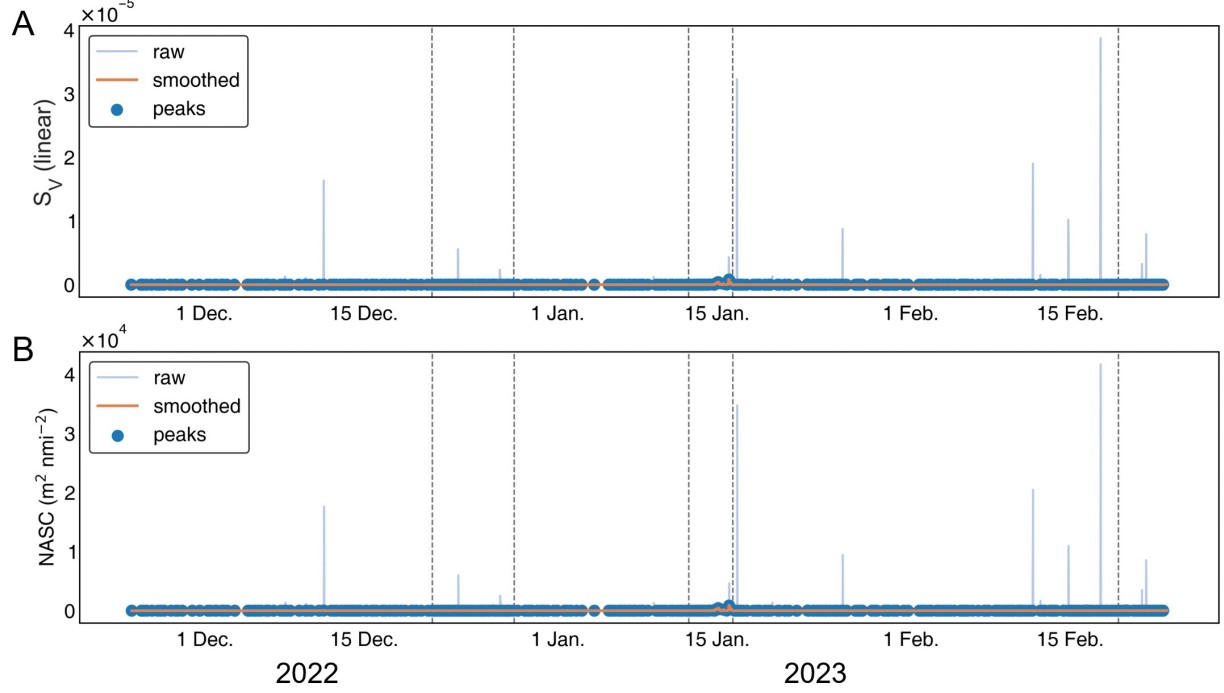

**Fig 3. Temporal variation in acoustic indices at the WBAT deployment station. (A)** $S_V$ (linear scale) and **(B)** NASC ($m^2$ $nmi^{-2}$). Blue lines show raw time-series values, and orange lines show smoothed data using a 5-point rolling median filter. Blue dots indicate detected local peaks. Vertical dashed lines denote changepoints identified by the PELT algorithm (rbf cost), indicating significant shifts in mean acoustic signal levels.

and 2,027,666 copies $mL^{-1}$ in the bottom layer were observed (Fig 6). This represents more than 1.5-fold (surface layer) and 3-fold (bottom layer) increases compared to the second-highest concentrations of 211,667 copies $mL^{-1}$ (surface) and 67,000 copies $mL^{-1}$ (bottom), recorded in late-January. After mid-January, eDNA concentrations gradually declined through late-February. By late-February and mid-March, *C. pallasii* eDNA was detected only at the surface (late-February) or only at the bottom (mid-March), indicating reduced and depth-variable presence during the end of the spawning season.

## Correlation among acoustic, catch, and eDNA

Spearman's rank correlation analysis (Fig 7) indicated moderate positive correlations between catch and eDNA in both the surface ($\rho = 0.56$) and bottom layers ($\rho = 0.48$). By contrast, $S_V$ was negatively correlated with catch ($\rho = -0.59$), suggesting that higher $S_V$ did not translate into increased catch over the survey period. A strong positive correlation was also observed between surface- and bottom-layer eDNA ($\rho = 0.85$), indicating consistent eDNA signal patterns across depths.

PCA results (Fig 8) showed that PC1 and PC2 captured a substantial proportion of the total variance, and samples were clearly separated by time period. December 2022 samples (15–16 December 2022; 27–28 December 2022) projected negatively on PC1, consistent with low catches and low eDNA concentrations. In contrast, January–February 2023 samples (12–13 January 2023; 30–31 January 2023; 15–16 February 2023) projected positively on PC1, indicating associations with higher catches and stronger eDNA signals. Notably, the 15–16 February 2023, sample exhibited elevated PC2 scores, suggesting variation along a secondary gradient associated with variation in $S_V$ relative to eDNA concentrations.

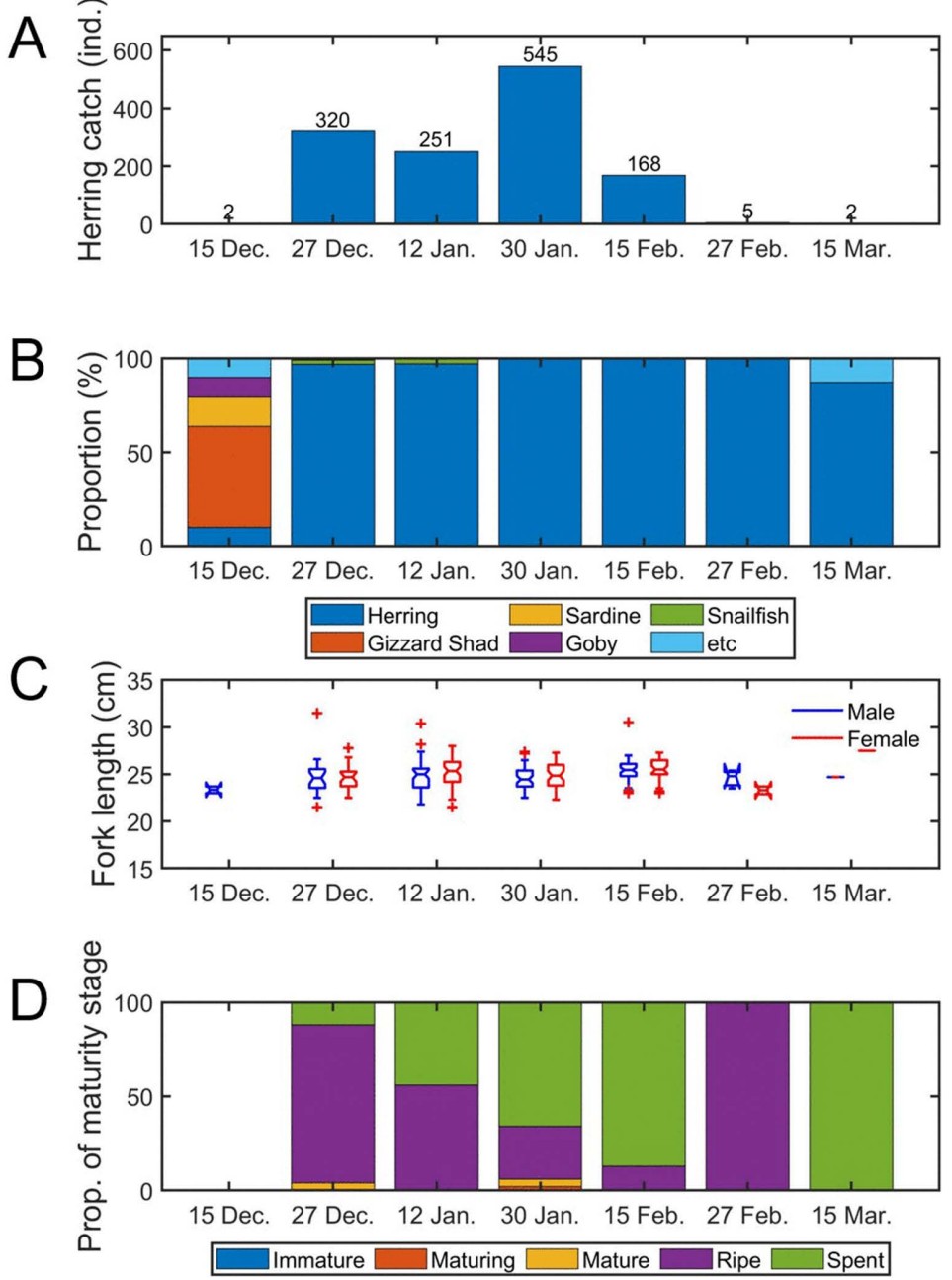

**Fig 4. Gillnet catch of Pacific herring (*Clupea pallasii*) in Jinhae Bay from December 2022 to March 2023. (A)** Number of herring captured during each survey, **(B)** Species composition (%) of total catch, **(C)** Fork length distribution of female and male herring, **(D)** Proportion of female maturity stages. Herring abundance peaked in late-January, coinciding with a high proportion of ripe and spent females.

## Discussion

In this study, we integrated a WBAT survey, gillnet survey, and eDNA analysis to continuously monitor the dynamics of *C. pallasii* in Jinhae Bay. Although acoustic observations provided continuous measurements of fish distribution and

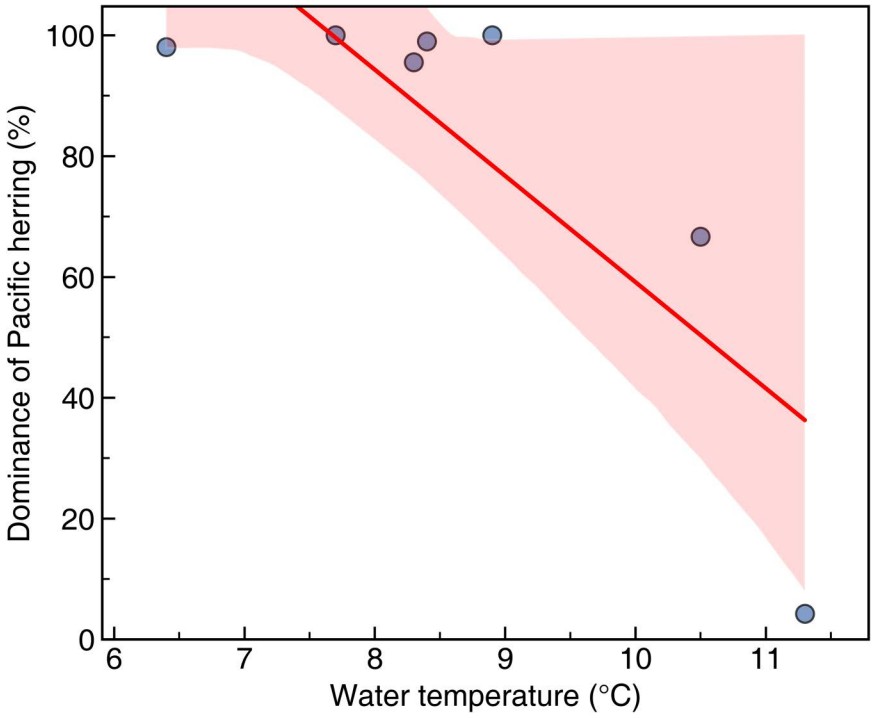

**Fig 5. Relationship between water temperature and dominance of Pacific herring (*Clupea pallasii*).** Dominance was calculated as the proportion of herring in the total gillnet catch for each survey. A significant negative association was observed, indicating increased herring dominance as water temperature decreased.

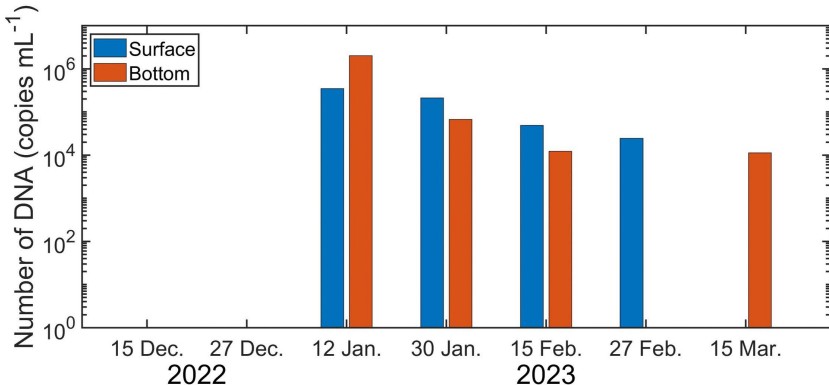

**Fig 6. Temporal variation in Pacific herring (*Clupea pallasii*) eDNA concentration in the surface and bottom waters of Jinhae Bay.** *C. pallasii* eDNA concentrations (copies mL$^{-1}$) were measured from December 2022 to March 2023 in both surface and bottom layers. eDNA was absent in December samples and in one of the layers during late-February and mid-March. Copy numbers peaked in mid-January, aligning with the main spawning influx observed in acoustic and catch data.

density, species identification required the complementary use of gillnet and eDNA data, thereby improving the reliability of interpretations. Specifically, gillnet surveys supplied detailed biological information, including species composition and maturity stages, while eDNA assays verified species occurrence in a non-invasive manner and supported temporal patterns detected acoustically. Together, these approaches offered a coherent and multidimensional assessment of herring

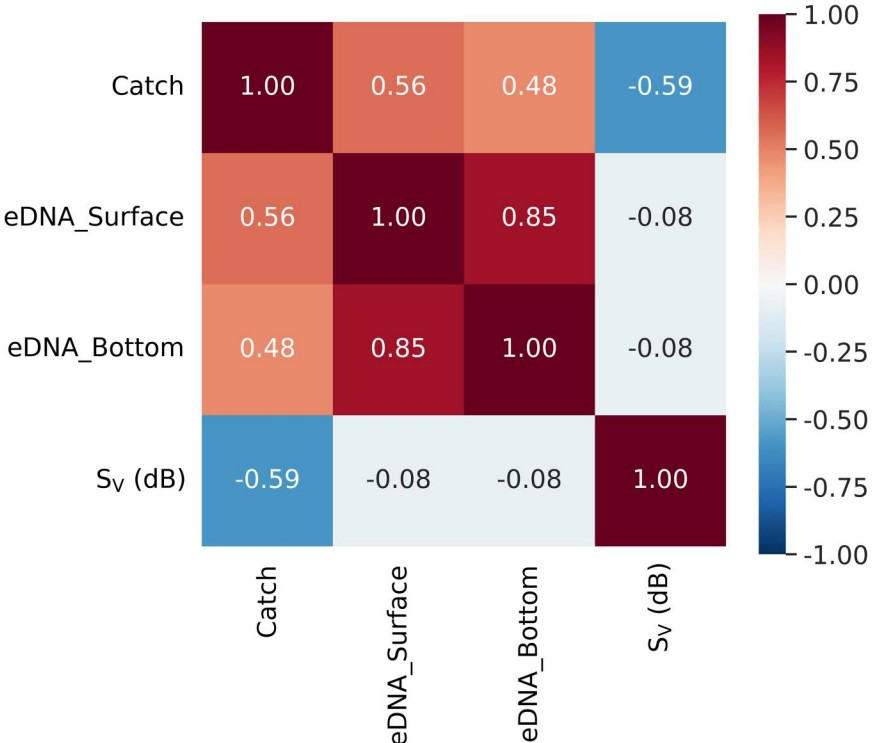

**Fig 7. Spearman's rank correlation matrix among acoustic backscatter ($S_v$), gillnet catch, and eDNA concentrations in surface and bottom waters.** The heatmap shows pairwise Spearman correlation coefficients ($\rho$) for the three survey datasets. Catch exhibited moderate positive correlations with both surface and bottom eDNA, whereas $S_v$ was negatively correlated with catch. Surface and bottom eDNA concentrations were strongly correlated, indicating consistent vertical distribution of eDNA signals.

migration, density, and spawning activity, strengthening the ecological interpretation of winter herring dynamics in Jinhae Bay. Jinhae Bay is a recognized winter spawning area for *C. pallasii*, where adults migrate to shallow coastal waters with abundant seaweed under weak-current conditions [11,26]. In this study, the WBAT was moored in coastal waters near Suchi Village in Jinhae Bay, a known spawning habitat, and acoustic detections, gillnet catches, and eDNA signals all converged to show the presence of spawning adults from late-December through mid-February. This timing aligns with previous submarine surveys documenting herring eggs in the same region from December to January [27], indicating persistent and repeated use of this spawning ground. Of the seven gillnet surveys conducted at the study site—excluding mid-December, late-February, and mid-March, when catch volume was low—*C. pallasii* was the clearly dominant species in the remaining four surveys, accounting for over 98% of the total catch volume (Fig 4B). In the initial mid-December survey, dotted gizzard shad and sardines were dominant, but dominance subsequently shifted to *C. pallasii*. Catch volume increased sharply from late-December and then declined after mid-February. All captured *C. pallasii* were adults (FL: 23.4–26.1 cm), mostly in the ripe or spent stages of reproductive maturity, indicating mass migration into these waters for spawning. At the Minedomari coast of Hokkaido, Japan, *C. pallasii* have been reported to visit the same waters annually for spawning [6], and a winter submarine survey off Tongyeong also repeatedly confirmed the presence of *C. pallasii* eggs [5]. Thus, the *C. pallasii* entering Jinhae Bay are therefore inferred to repeatedly use the same spawning grounds.

Considering the environmental drivers of this annual migration, water temperature was a key factor associated with herring presence. Herring first appeared when bottom temperatures fell below 10 °C, and the highest catch occurred when bottom temperatures averaged 6.72 °C in January. This range is consistent with spawning temperatures (5–8 °C) reported

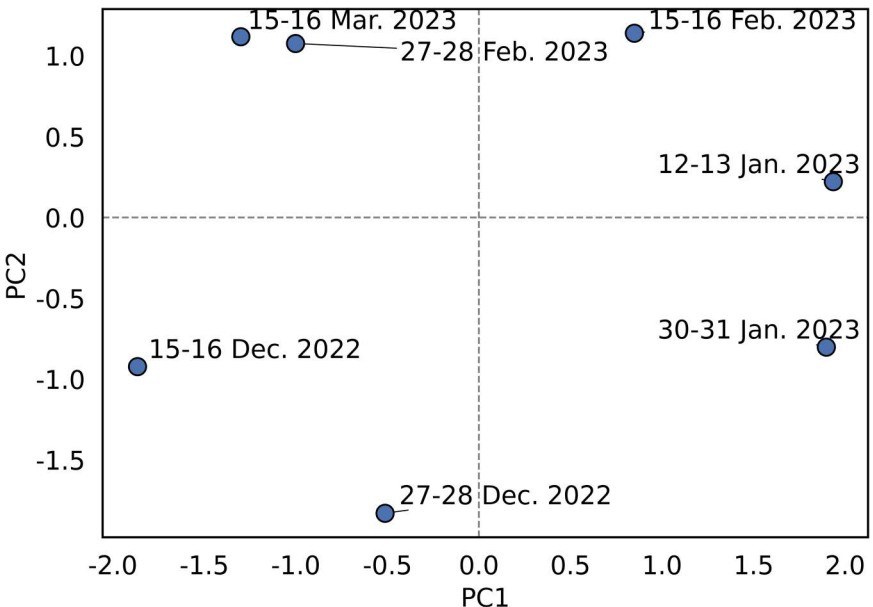

**Fig 8. Principal component analysis (PCA) of acoustic backscatter (S$_v$), gillnet catch, and eDNA concentrations in surface and bottom waters in Jinhae Bay.** Sample points are labeled by survey date. PC1 and PC2 summarize major temporal changes in herring occurrence across surveys. December samples grouped negatively on PC1, reflecting low catch and low eDNA levels, whereas January-February samples clustered positively on PC1, corresponding to peak spawning activity and high biological signal intensity.

in Tongyeong and Jinhae Bay [26], as well as spawning temperatures observed in Hokkaido and British Columbia [28,29]. Globally, *C. pallasii* exhibit a wide latitudinal gradient in spawning timing, from October in California to July in Alaska [30]. The timing observed in this study, from late-December to mid-February, aligns with the broader seasonal pattern reported for northwest Pacific herring populations. These findings suggest that seasonal cooling serves as a strong environmental cue triggering herring migration into Jinhae Bay.

Using a WBAT moored for approximately 90 days, we identified distinct diel patterns in herring behavior. Throughout winter, fish school signals were detected irregularly, appearing both as densely concentrated schools and as broadly dispersed layers. A pronounced and continuous increase in acoustic signal strength occurred from 14–17 January 2023, and detailed examination of echograms from this period showed that *C. pallasii* were vertically dispersed across the water column rather than forming compact, high-density aggregations (Fig 9A). In January, when the survey was conducted—which is winter in South Korea—sunrise and sunset times were at 07:34 and 17:38 local time, respectively. The *C. pallasii* school signals were detected more at night than during the daytime (Fig 9B), and strong echo signals were observed in the middle-to-bottom layers at night. The mean S$_v$ was −62.99 dB during the day and −54.90 dB during the night, meaning that the S$_v$ was more than 8 dB higher at night. Similar diel behavior has been observed in previous studies. In Miyako Bay, Japan, ultrasonic telemetry revealed that adult herring remain in deeper waters during the day and migrate shoreward at night to spawn [31]. Experiments with tagged herring in Hokkaido also showed that spawning activity predominantly occurred at night, particularly after ~20:20 h [32]. Atlantic herring in Norway form horizontally broad, loosely packed layers near the seabed shortly after sunset, coinciding with intense nocturnal spawning activity [33]. In British Columbia, spawning timing and spatial patterns of *C. pallasii* have been shown to depend on day length and cumulative thermal experience, with repeated spawning activity occurring in the same regions [29]. Taken together, these findings align with our observations and strongly suggest that the dispersed nighttime formations and heightened nighttime S$_v$ in Jinhae Bay reflect the typical nocturnal spawning behavior of *C. pallasii*. When these acoustic patterns are interpreted alongside

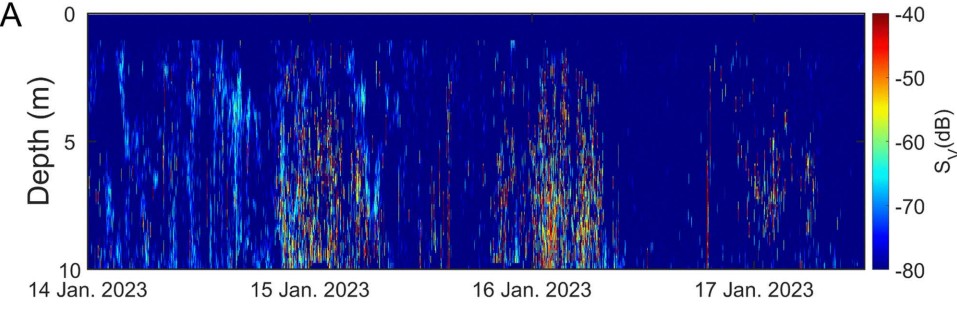

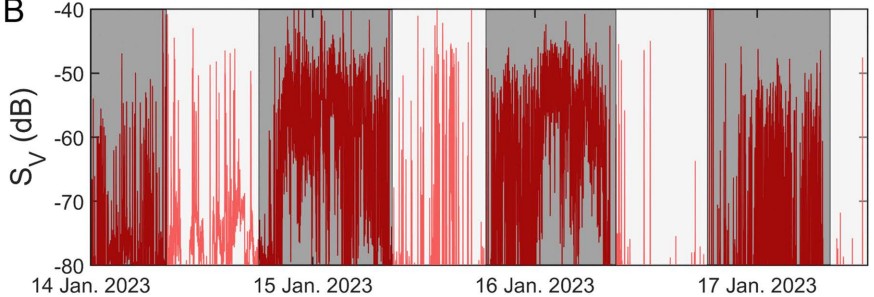

**Fig 9. Echogram and diel variation of Pacific herring acoustic backscattering during the peak spawning influx from 14–17 January 2023. (A)** Echogram of 70 kHz backscatter showing vertically dispersed herring signals throughout the water column during the peak influx period. **(B)** Diel pattern of mean $S_V$. White shading indicates daytime (06:00-19:00) and gray shading indicates nighttime (19:00-06:00). Nighttime $S_V$ was consistently higher than daytime $S_V$, demonstrating nocturnal aggregation behavior associated with spawning activity.

gillnet catch results and maturity-stage data, they indicate a clear and concentrated influx of spawning adults during mid-winter. To determine whether this biological pattern was also reflected in the surrounding water column, we next examined temporal changes in eDNA concentration.

Meanwhile, continuous WBAT monitoring showed that fish school signals appeared intermittently, and during the three nights from 14–17 January 2023, fish were detected not compact schools but as strong, vertically dispersed layers throughout the entire water column (Fig 9). Due to the shallow depth (~11 m) at the deployment site and the narrow beam width (< 7°) of the WBAT, the instrument could not fully cover all potential spawning grounds; however, the observations were still sufficient to identify the primary period of spawning migration of *C. pallasii* into the area. Taken together, these acoustic patterns, along with the catch volume, size structure, and maturity stages, confirm a pronounced influx of spawning adults in mid-winter. To determine whether this biological signal was also reflected in the surrounding water column, we next examined temporal variation in eDNA concentration. The WBAT was originally planned to operate until 31 March 2023, to determine the precise timing of *C. pallasii* entry into and exit from Jinhae Bay, as well as the movements of larvae and juveniles following spawning. However, the survey was terminated in late-February due to a battery failure, preventing observation of the period during which juveniles typically exit the bay. A previous WBAT deployment in Jinhae Bay detected signals presumed to be juveniles from early to late-March [13]. Because *C. pallasii* generally hatch approximately 12 days after fertilization [11], future deployments will extend operations through April to enable continuous monitoring of interannual variability in the timing of larval and juvenile outmigration. In the qPCR-based eDNA analysis in our study, although contamination was observed in an NC sample collected at sea on 12 January 2023, this is thought to be due to cross-contamination of the NC sample with *C. pallasii* DNA on site, despite the use of thoroughly sterilized collection bottles. This cross-contamination was only observed in this on-site NC, and there was no contamination of other NCs, including the lab NCs. Therefore, although some caution is required when interpreting the quantitative concentrations from

the 12 January sample, it is clear that *C. pallasii* was present in the waters from this time. In late-January, mid-February, and late-February, we observed a higher eDNA concentration in the surface layer than the bottom layer. It is thought that, because the main spawning depth for *C. pallasii* in Jinhae Bay is typically within 10 m, large quantities of DNA released during spawning will have been more concentrated in the surface layer [27]. The increase in eDNA during spawning activity is consistent with previous reports of a steep increase in eDNA concentration during spawning activity by sockeye salmon (*Oncorhynchus nerka*) and an increase in eDNA concentration during spawning and fertilization by Japanese eel (*Anguilla japonica*) in a breeding experiment [34,35]. On the other hand, in mid-March, *C. pallasii* eDNA was only detected in the bottom layer. This is because, after the end of spawning activities, eggs that had sunk or were attached to the sea-bed, including reefs, were present in the bottom layer, and so eDNA released from the eggs was detected in samples from the bottom layer. Thus, these results reflect the ecological characteristics of *C. pallasii* [5].

Looking forward, year-round WBAT moorings combined with standardized gillnet and eDNA monitoring would enable assessment of interannual variation in migration timing, spawning intensity, and juvenile departure. Such integrated surveys could help identify environmental drivers—including temperature, salinity, and oxygen—that structure seasonal herring dynamics. Expanding this monitoring framework to other pelagic species (e.g., horse mackerel, anchovy, Pacific cod) would allow ecosystem-level evaluation of overlapping spawning habitats and support comprehensive stock assessment. This evidence base can inform management actions such as spawning-ground protection zones, seasonal closures, and gear-size regulations to minimize harvesting of immature fish. Ultimately, incorporating environmental data and population indicators into a real-time resource-monitoring system would support adaptive, climate-ready management of coastal fish resources.

## Conclusion

This study integrated long-term acoustic monitoring, gillnet surveys, and eDNA analysis to clearly delineate the migration period and spawning patterns of *C. pallasii* entering Jinhae Bay, South Korea. All three datasets consistently indicated that herring density began increasing in late-December, peaked in January, and declined after late-February. The strong agreement among acoustic, biological, and molecular indicators provides a reliable basis for defining the winter spawning dynamics of herring in this region.

These findings offer fundamental information for *C. pallasii* resource management and sustainable fisheries operations. In particular, spawning season catch restrictions and spawning ground protection zones are suggested to be required, taking into account the peak spawning season and key spawning areas. Furthermore, catch limits based on gear type and size are recommended to be implemented to prevent overfishing of immature individuals. In addition, by combining long-term acoustic observations with eDNA monitoring, a real-time resource fluctuation monitoring system can be established, through which resource management may be adapted rapidly to reflect changes in the *C. pallasii* population associated with environmental variability.

## Acknowledgments

The authors sincerely thank Professor Woo-seok Gwak (Gyeongsang National University) for his kind cooperation and support in the analysis of environmental DNA samples.

## Author contributions

**Conceptualization:** Euna Yoon, Hyungbeen Lee.

**Formal analysis:** Inyeong Kwon, Hyungbeen Lee.

**Investigation:** Euna Yoon, Yong-Deuk Lee, Jeong-Hoon Lee, Hyungbeen Lee.

**Methodology:** Euna Yoon, Hyungbeen Lee.

**Project administration:** Jeong-Hoon Lee.

**Software:** Euna Yoon, Hyungbeen Lee.

**Validation:** Euna Yoon.

**Writing – original draft:** Euna Yoon, Inyeong Kwon, Hyungbeen Lee.

**Writing – review & editing:** Euna Yoon, Yong-Deuk Lee, Jeong-Hoon Lee, Inyeong Kwon, Hyungbeen Lee.

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
