## [Decision Letter · Decision Letter 0]

13 Aug 2025

Dear Dr. Lee,

Thank you for submitting your manuscript to PLOS ONE. After careful consideration, we feel that it has merit but does not fully meet PLOS ONE’s publication criteria as it currently stands. Therefore, we invite you to submit a revised version of the manuscript that addresses the points raised during the review process.

We look forward to receiving your revised manuscript.

Kind regards,

Mizanur Rahman, Ph.D.

Academic Editor

PLOS ONE

Journal Requirements:

2. To comply with PLOS ONE submissions requirements, in your Methods section, please provide additional information regarding the experiments involving animals and ensure you have included details on (1) methods of sacrifice, and (2) efforts to alleviate suffering.

5. In the online submission form, you indicated that the data can be shared upon request by contacting the corresponding author.

7. We note that the grant information you provided in the ‘Funding Information’ and ‘Financial Disclosure’ sections do not match.

8. Thank you for stating the following financial disclosure:

This work was supported by the National Institute of Fisheries Science, Korea (grant No. R2025001).

9. We note that Figure 1 in your submission contain [map/satellite] images which may be copyrighted. All PLOS content is published under the Creative Commons Attribution License (CC BY 4.0), which means that the manuscript, images, and Supporting Information files will be freely available online, and any third party is permitted to access, download, copy, distribute, and use these materials in any way, even commercially, with proper attribution. For these reasons, we cannot publish previously copyrighted maps or satellite images created using proprietary data, such as Google software (Google Maps, Street View, and Earth). For more information, see our copyright guidelines: http://journals.plos.org/plosone/s/licenses-and-copyright.

Additional Editor Comments :

Major concern

1. Require Methodological Detail for Replication: Key parts of the methods lack the specificity needed for reproducibility (e.g., eDNA sampling volumes, filter pore size, type of preservation buffer, extraction kit details, qPCR primer sequences, calibration procedures for acoustic surveys). Expand the methods to include all equipment models, manufacturer names, software versions, parameter values, and QA/QC procedures.

2. Limited Integration of Multi-Method Results: Results from WBAT acoustic surveys, gillnet sampling, and eDNA analysis are largely reported separately with minimal cross-comparison. Add a synthesis subsection that directly compares detection rates, abundance patterns, and spatial overlap between methods, supported by statistics.

3. Statistical Rigor in Data Interpretation is required: Some conclusions are based on qualitative observations or descriptive statistics without formal hypothesis testing (e.g., seasonal patterns, species distributions). Apply appropriate statistical analyses (ANOVA, regression, correlation tests) to quantify differences and report effect sizes and p-values.

4. Link Between Findings and Management Recommendations needs more concise: While the discussion mentions potential management measures, these are vague and not clearly tied to the specific results obtained. Use the data to justify concrete, actionable recommendations—e.g., define spatial closure boundaries based on detection hotspots, specify timing of closures from seasonal abundance peaks.

Minor concern

Line(s) Comment

L2–4 Remove extra spaces before punctuation/parentheses and italicize species names: “Clupea pallasii”), monitoring, gillnet…”

L40–48 Correct spacing issues ('mid -January' → 'mid‑January', 'In summa ry' → 'In summary') and add recent citations.

L88–95 Add coordinates for study sites, a brief oceanographic description, and a map reference.

L96–119 Report full acoustic survey parameters (transducer depth, beam angle, pulse length, calibration method, TS thresholds).

L136–156 Include eDNA filtration volume, filter type, field/extraction blanks, qPCR efficiency, R², LOD/LOQ, and inhibition tests.

L165–176 Provide NASC summary statistics with confidence intervals; ensure figure axes are labeled with units.

L200–204 Add sample sizes (n) for maturity stage percentages.

L210–226 Report standard curve details for qPCR and include contamination control measures.

L227–238 Synthesize results across WBAT, gillnet, and eDNA instead of restating them.

L258–272 Propose clear temporal closure and spatial buffer recommendations for fisheries management.

L313–349 Add more detailed future research directions, e.g., egg mapping, larval surveys, habitat covariates.

L350–358 Add an actionable management takeaway in the conclusion.

L385–460 Standardize reference formatting, fix spacing, and add DOIs where available.

Reviewers' comments:

Reviewer's Responses to Questions

**Comments to the Author**

1. Is the manuscript technically sound, and do the data support the conclusions?

Reviewer #1: Yes

2. Has the statistical analysis been performed appropriately and rigorously?

Reviewer #1: No

3. Have the authors made all data underlying the findings in their manuscript fully available?

Reviewer #1: No

4. Is the manuscript presented in an intelligible fashion and written in standard English?

Reviewer #1: Yes

Reviewer #1: General comments

The authors may benefit from language editing to enhance the clarity of their work and reduce ambiguity. A lot of the writing uses active voice, especially in the materials and methods, results and discussion sections of the work. The combination of acoustic telemetry, gill net surveys and eDNA is indeed interesting since these complementary methods allow for the triangulation of spawning events with greater precision than just using one approach. As such, it brings out the spatial and temporal dynamics of C. pallasii spawning. The study further lacks details in the methodology section thus impacting on reproducibility of the work.

The results for this study lacks statistical analyses to tease out any temporal differences in spawning activity and whether correlations exist between the different methods. Authors should consider adding appropriate statistical analyses to investigate differences, if any. For the discussion, the authors may consider revising and improving the discussion for clarity and flow from one paragraph to another. Finally, it would be great if the authors could demonstrate how study improves our knowledge on spawning ecology and how it could possibly be used to aid fisheries monitoring.

Specific comments

Introduction

Line 60 – 61: The following sentence is unclear:

“To protect C. pallasii resources, which are commercially and biologically important, since 2022, South Korea has implemented a minimum landing size of 20 cm [10].”

The authors may consider rephrasing to enhance clarity. “Resources” here is a bit vague.

Line 67-69: Similarly, the following sentence is also unclear:

“On the other hand, for accurate species identification, other survey methods need to be used in parallel, such as a gillnet catch survey or eDNA analysis.”

Also consider revising for clarity.

Line 79-86: While the aim is “…ascertaining fluctuations in C. pallasii occurrence and comprehensively analysing spawning season characteristics” and the objective is “…to ascertain the time and relatively volume of C. pallasii migration to Jinhae Bay for spawning,” the primary objective is not coming out clearly. The authors may consider rephrasing to highlight what the primary objective is since it is not clear whether the study is focused on only the spawning season(s) or if migration patterns are important as part of the spawning season.

Materials and Methods

Authors should consider using passive voice in the materials and methods section instead of active voice. All through the text, there was a mix of active and passive voice. For example, they should avoid using or starting with “To monitor…” (line 90), “To install…” (line 100) and re-write the sentences in passive voice.

Overall, this section needs revision to improve clarity.

Lines 90 – 94:

Firstly, the study lacks description. Authors to consider including a brief description of the study site before providing the sampling schedule.

Secondly, authors may consider having figure 1 appearing before table 1 under the Study Site and Schedule subheading, prior to WBAT installation and analysis of acoustic data subheading. Then Figure 1 could be referred to from within the proceeding subheading.

Authors should state how many stations for the study. Table 1 caption/title should be re-written for clarity.

Lines 99 – 125: Authors should consider re-writing in passive voice for clarity. While depth of the many sites have been mentioned in Table 1, it may be good for the authors to mention it in the WBAT installation process and for WBAT, Provide the full names of what it actually is then continue using abreviations. Furthermore, what parameters do the WBAT record? Please provide this in the text.

What was the mooring period for the installed WBAT, i.e., for how long were the WBAT installed for? This is missing.

Lines 127 – 135: Authors should consider re-writing this section in passive voice for clarity. While gear specifications have been mentioned, there lacks clarity.

For instance, what was the survey period, season, sampling frequency (e.g., daily, weekly, monthly, etc), environment (e.g., habitat type, etc)?

How were the gears deployed and are they standardized?

What was the total number of fish collected from the gill net survey? i.e., How was the catch handled (e.g., total number of fish caught/sampled, species identification as per reference guides, length, weight, sex and maturity stage)?

Lines 138 - 155: Authors should consider re-writing this section in passive voice for clarity. Authors should avoid using the active voice “To analyse C. pallasii eDNA …”and revise to passive voice. This appears in several instances in the sub-section. The sampling design for water sampling collection for eDNA is not clear.

Authors should clarify surface layer depth (e.g. is it 0 – 1 m?), bottom depth whether for each station water is collected (e.g. is it 8 m, 10 m, or 12 m, etc) or a single bottom depth if it was uniform across all stations.

For eDNA extraction, while QIAGEN DNeasy Blood and Tissue Kits were used, what was the protocol for extraction? Were manufacturer’s instructions followed or modified? Were there any blanks or control used during DNA extraction? In addition, the reasons for DNA extraction to compare to eDNA samples is not clear.

For PCR amplification, there is no mention of the primers used/developed, and if developed, how they were developed. There is also no mention of the thermal cycling conditions. Consider including all these.

Are there any statistical analysis conducted to identify peak spawning periods apart from the time series analysis �e.g. comparing seasonal/monthly differences in catch rates or eDNA signals�and was consistency between the different methods validate (e.g., is there any correlation between eDNA and WBAT acoustic, etc)? Authors should consider adding appropriate statistical analyses to investigate differences, if any.

Results

Authors should consider revising using passive voice for clarity and go directly to focussing on just the results. Also, considering that statistical analyses was not mentioned in the methods, are there any temporal differences in catch and size, differences in species and maturity stage proportions or eDNA signals that reflect spawning activity?

Lines 159 – 160: While water temperature has been mentioned as part of the results, authors have not mentioned how they arrived to this in the methods section.

Lines 211: Authors mentioned the presence of contamination in their eDNA results yet in line 157 of methods, they did mention on how analysis was conducted to prevent contamination. Why would there be contamination in the NC sample from mid-January but not in other NC samples? What was the role of the DNA extraction from tissue samples of the C. pallasii?

Discussion

While the discussion has compared the study results for the different methods used to assess Winter occurrence and spawning characteristics of Pacific herring with other studies, the paragraphs appear seem disjointed from one to the next. For instance, in 2nd paragraph, the authors discuss the 7 gillnet surveys but then discuss the again in the 6th paragraph right after discussion of the qPCR eDNA analysis results. Same instance can be said on WBAT discussion (4th and 7th paragraphs).

The authors may consider using the last paragraph of the discussion as the introducing paragraph of the Discussion since it provides a quick snapshot of the results before the results are discussed at length in the proceeding paragraphs.

Conclusion

The authors should consider writing the conclusion in a single paragraph with just the key findings, emphasising on how the benefits of using the complementary methods approach and any future applications or research where it can be used. Were there any limitations in the study? What are the future directions? How does this study or can this study provide more knowledge on spawning ecology and fisheries monitoring?

**Do you want your identity to be public for this peer review?** For information about this choice, including consent withdrawal, please see our Privacy Policy

Reviewer #1: No

---

## [Author Response · Author response to Decision Letter 1]

29 Oct 2025

Dear Editor,

Thank you for evaluating our manuscript; we appreciate the time you and the Reviewers have taken to provide guidance and thoughtful suggestions from which the manuscript has benefited. We have revised the manuscript accordingly. Please find our point-by-point responses to all comments below.

[Editor 1]

2. To comply with PLOS ONE submissions requirements, in your Methods section, please provide additional information regarding the experiments involving animals and ensure you have included details on (1) methods of sacrifice, and (2) efforts to alleviate suffering.

We sincerely thank the editor for the comment. All procedures were conducted in strict accordance with the recommendations provided in the Guide for the Care and Use of the National Institutes of Fisheries Science and followed the protocol approved by the Committee on the Ethics of the National Institutes of Fisheries Science (Protocol Number: 2022-NIFS-IACUC-24).

We sincerely thank the editor for the comment. All procedures were conducted in strict accordance with the recommendations provided in the Guide for the Care and Use of the National Institutes of Fisheries Science and followed the protocol approved by the Committee on the Ethics of the National Institutes of Fisheries Science (Protocol Number: 2022-NIFS-IACUC-24).

We sincerely thank the editor for the comment. The datasets analyzed during the current study were obtained from national sources and are the property of the National Institute of Fisheries Science. Due to legal and institutional restrictions, public sharing of these datasets is not permitted. However, the data can be made available upon reasonable request in a “Contact for Data Access” format. Researchers interested in accessing the data may contact the corresponding author for consideration. Other researchers are currently preparing similar datasets through different means.

5. In the online submission form, you indicated that the data can be shared upon request by contacting the corresponding author.

We sincerely thank the editor for the comment. As indicated in the online submission form, the datasets analyzed during the current study are the property of the National Institute of Fisheries Science and cannot be publicly shared due to legal and institutional restrictions. However, the data will be provided upon reasonable request in a “Contact for Data Access” format by contacting the corresponding author.

We sincerely thank the editor for the comment. We have revised the data availability statement to clarify that the datasets analyzed during this study are the property of the National Institute of Fisheries Science and cannot be publicly shared due to legal and institutional restrictions. The data will only be provided if necessary, upon reasonable request, in a “Contact for Data Access” format by contacting the corresponding author.

7. We note that the grant information you provided in the ‘Funding Information’ and ‘Financial Disclosure’ sections do not match.

We sincerely thank the editor for pointing this out. The Funding Information and Financial Disclosure sections have been corrected to consistently state: This work was supported by the National Institute of Fisheries Science, Korea (grant No. R2025001).

8. Thank you for stating the following financial disclosure:

This work was supported by the National Institute of Fisheries Science, Korea (grant No. R2025001).

We sincerely thank the editor for the comment. The funders provided support for this work and were involved in study design, data collection and analysis, the decision to publish, and preparation of the manuscript.

9. We note that Figure 1 in your submission contain [map/satellite] images which may be copyrighted. All PLOS content is published under the Creative Commons Attribution License (CC BY 4.0), which means that the manuscript, images, and Supporting Information files will be freely available online, and any third party is permitted to access, download, copy, distribute, and use these materials in any way, even commercially, with proper attribution. For these reasons, we cannot publish previously copyrighted maps or satellite images created using proprietary data, such as Google software (Google Maps, Street View, and Earth). For more information, see our copyright guidelines: http://journals.plos.org/plosone/s/licenses-and-copyright.

We sincerely thank the editor for the comment. The map in Figure 1 was generated using the MATLAB M_Map toolbox [20] and therefore does not rely on copyrighted third-party map or satellite data.

We appreciate the editor suggestions. We have reviewed the recommended works and included the relevant references in the revised manuscript.

General Comments

1. Require Methodological Detail for Replication: Key parts of the methods lack the specificity needed for reproducibility (e.g., eDNA sampling volumes, filter pore size, type of preservation buffer, extraction kit details, qPCR primer sequences, calibration procedures for acoustic surveys). Expand the methods to include all equipment models, manufacturer names, software versions, parameter values, and QA/QC procedures.

We sincerely thank the reviewer for this valuable comment. In the revised manuscript, we have expanded the Methods section to enhance reproducibility (P9L158-P10L187). Specifically, we have now included:

- eDNA sampling volume (2 L from both surface and bottom layers)

- Filter pore size (0.45 μm, Sterivex, Millipore, USA)

- Preservation buffer (RNAlater, Thermo Fisher Scientific, USA)

- DNA extraction kit (DNeasy Blood and Tissue Kit, Qiagen, Germany)

- DNA quantification method (Synergy H1, BioTek, USA with QuantiFluor dsDNA System, Promega, WI, USA)

- qPCR primer and probe sequences targeting the C. pallasii cytochrome b region (Gwak and Nakayama 2021, Conservation Genetics Resources 13: 337-339)

Furthermore, we have clarified QA/QC procedures, including negative controls at both laboratory and qPCR steps and triplicate analyses, and listed the manufacturer names and model numbers for all instruments. These revisions ensure that the methodology is fully transparent and reproducible.

2. Limited Integration of Multi-Method Results: Results from WBAT acoustic surveys, gillnet sampling, and eDNA analysis are largely reported separately with minimal cross-comparison. Add a synthesis subsection that directly compares detection rates, abundance patterns, and spatial overlap between methods, supported by statistics.

We sincerely thank the reviewer for the valuable suggestion. In the revised manuscript, we have added a new synthesis subsection that directly compares detection rates, abundance patterns, and spatial overlap among WBAT acoustic surveys, gillnet sampling, and eDNA analysis. This synthesis integrates results across methods and is supported by appropriate statistical analyses, providing a clearer, multi-method perspective on species occurrence and distribution (P15L307-P16L328).

3. Statistical Rigor in Data Interpretation is required: Some conclusions are based on qualitative observations or descriptive statistics without formal hypothesis testing (e.g., seasonal patterns, species distributions). Apply appropriate statistical analyses (ANOVA, regression, correlation tests) to quantify differences and report effect sizes and p-values.

We sincerely thank the reviewer for the helpful comment. In the revised manuscript, we have added a dedicated paragraph in the Materials and Methods section under “Statistical Analysis” (P10L189-P11L219) and in the Results section under “Correlation among acoustic, catch, and eDNA” to quantitatively evaluate the relationships among these datasets(P15L307-P16L328). Additionally, the Discussion has been updated to include interpretation of these correlations, providing a more comprehensive understanding of the observed patterns.

4. Link Between Findings and Management Recommendations needs more concise: While the discussion mentions potential management measures, these are vague and not clearly tied to the specific results obtained. Use the data to justify concrete, actionable recommendations—e.g., define spatial closure boundaries based on detection hotspots, specify timing of closures from seasonal abundance peaks.

We sincerely thank the reviewer for this valuable suggestion. The final part of the Discussion and the Conclusion have been revised to provide more concrete and actionable management recommendations(P22L447-P22L465).

Specific comment

L2–4 Remove extra spaces before punctuation/parentheses and italicize species names: “Clupea pallasii”), monitoring, gillnet…”

We thank the reviewer for the helpful comment. The extra spaces have been removed, and the species names have been italicized accordingly (P1, L1–2).

L40–48 Correct spacing issues ('mid -January' → 'mid‑January', 'In summa ry' → 'In summary') and add recent citations.

We appreciate the reviewer’s suggestion. The spacing issues have been corrected, and the recent citations have been added as recommended.

L88–95 Add coordinates for study sites, a brief oceanographic description, and a map reference.

We appreciate the reviewer’s comment. Coordinates for the study sites, a brief oceanographic description, and a map reference have been added (P5L89-92). The map source has been included in the caption of Figure 1(P5L97-102).

L96–119 Report full acoustic survey parameters (transducer depth, beam angle, pulse length, calibration method, TS thresholds).

The requested acoustic survey parameters have been added to the manuscript (P7 L117-130).

L136–156 Include eDNA filtration volume, filter type, field/extraction blanks, qPCR efficiency, R², LOD/LOQ, and inhibition tests.

We appreciate the reviewer’s suggestion. The section on “eDNA survey for species identification” has been revised to include the requested details.

L165–176 Provide NASC summary statistics with confidence intervals; ensure figure axes are labeled with units.

We thank the reviewer for this helpful suggestion. In response, we have revised the Results and figure descriptions as follows: NASC summary statistics cannot be presented with confidence intervals because the values are extracted as mean estimates at the time of data processing. Figure labeling: We have ensured that all axes in Fig. 2 are labeled with appropriate units. Specifically, NASC values are expressed in m² nmi⁻², and volume backscattering strength (Sv) is provided in dB. Figure legend update: The Fig. 2 legend has been revised to clarify the definitions of each panel, including NASC as the integrated backscattering normalized over horizontal area (m² nmi⁻²) (P12L245–P13L250).

L200–204 Add sample sizes (n) for maturity stage percentages.

We thank the reviewer for the comment. Sample sizes (n) for maturity stage percentages have been added (P14L274-P14L279).

L210–226 Report standard curve details for qPCR and include contamination control measures.

We thank the reviewer for the comment. Contamination prevention procedures, including equipment sterilization and the use of negative controls (field NC, Lab NC, and qPCR NC), were implemented throughout the study to monitor potential contamination.

L227–238 Synthesize results across WBAT, gillnet, and eDNA instead of restating them.

We sincerely thank the reviewer for the comment. The first paragraph of the Discussion has been revised to synthesize results across WBAT, gillnet, and eDNA, rather than restating them (P15L307-P16 L325).

L258–272 Propose clear temporal closure and spatial buffer recommendations for fisheries management.

We sincerely thank the reviewer for this valuable suggestion. In response, we have revised the Discussion and Conclusion sections to explicitly address both temporal and spatial management measures. Specifically, we now (i) propose spawning season catch restrictions that take into account the peak spawning period of C. pallasii as a form of temporal closure and (ii) recommend the designation of spawning ground protection zones together with adjacent spatial buffer areas to reduce fishing pressure in critical habitats. In addition, gear-based management measures (e.g., adjustments to gillnet mesh size) and the establishment of a real-time monitoring system integrating acoustic and eDNA observations have been included to further strengthen evidence-based fisheries management recommendations.

L313–349 Add more detailed future research directions, e.g., egg mapping, larval surveys, habitat covariates.

We sincerely thank the reviewer for the suggestion. A new paragraph outlining detailed future research directions has been added accordingly (P22L447-P22L465).

L350–358 Add an actionable management takeaway in the conclusion.

We sincerely thank the reviewer for the comment. The Conclusion has been thoroughly revised to include an actionable management takeaway.

[Reviewer 1]

General comments

The authors may benefit from language editing to enhance the clarity of their work and reduce ambiguity. A lot of the writing uses active voice, especially in the materials and methods, results and discussion sections of the work. The combination of acoustic telemetry, gill net surveys and eDNA is indeed interesting since these complementary methods allow for the triangulation of spawning events with greater precision than just using one approach. As such, it brings out the spatial and temporal dynamics of C. pallasii spawning. The study further lacks details in the methodology section thus impacting on repro

---

## [Editor Report · Decision Letter 1]

25 Nov 2025

Winter occurrence and spawning characteristics of Pacific herring (Clupea pallasii) in Jinhae Bay: An integrated survey using acoustic monitoring, gillnet sampling, and environmental DNA

PONE-D-25-36896R1

Dear Dr. Lee,

We’re pleased to inform you that your manuscript has been judged scientifically suitable for publication and will be formally accepted for publication once it meets all outstanding technical requirements.

Kind regards,

Mizanur Rahman, Ph.D.

Academic Editor

PLOS ONE

Additional Editor Comments (optional):

I congratulate you for the improvement of the manuscript. Although the manuscript is accepted for the time being (to avoid another round of revision), please try to improve the manuscript when you receive the proof copy

1. Clarity and flow: A few sentences in the Introduction and Discussion could be refined for smoother flow. Consider merging shorter sentences to improve readability, especially when transitioning between acoustic, gillnet, and eDNA results.

2. Consistency of terminology: Please double-check the manuscript for consistent use of technical terms (WBAT, Sv, NASC, eDNA copy number). Consistency will improve overall clarity for readers unfamiliar with multi-method survey approaches.

3. Minor formatting issues: Ensure all scientific names (like, Clupea pallasii) are italicized throughout. Some occurrences may still remain unitalicized. Check spacing around parentheses, hyphens, and units for full consistency.

4. Figures and legends: Figures are clear and informative; however, some legends could briefly restate what the key message of the figure is (for instance, “peak detection observed in mid-January”). This will help readers quickly understand the significance without referring back to Results.

5. Methodological clarity: Although significantly improved, a few method steps (especially statistical descriptions such as PCA preprocessing) may benefit from one or two clarifying sentences. This would ensure reproducibility for readers.

6. Discussion refinement: The Discussion is strong, but in a few places transitions between paragraphs could be made smoother (for example, when moving from gillnet patterns to eDNA dynamics. A brief bridging sentence may help maintain logical flow).
---

## [Editor Report · Acceptance letter]

PONE-D-25-36896R1

PLOS One

Dear Dr. Lee,

I'm pleased to inform you that your manuscript has been deemed suitable for publication in PLOS One. Congratulations! Your manuscript is now being handed over to our production team.

Kind regards,

on behalf of

Dr. Mizanur Rahman

Academic Editor

PLOS One